# Silicon Alleviates Copper Toxicity in Flax Plants by Up-Regulating Antioxidant Defense and Secondary Metabolites and Decreasing Oxidative Damage

**Hossam S. El-Beltagi** [1,2,*] , **Mahmoud R. Sofy** [3,*] , **Mohammed I. Aldaej** [1] **and Heba I. Mohamed** [4]

1   Agricultural Biotechnology Department, College of Agriculture and Food Sciences, King Faisal University, Al-Ahsa 31982, Saudi Arabia; maldaej@kfu.edu.sa
2   Biochemistry Department, Faculty of Agriculture, Cairo University, Giza 12613, Egypt
3   Botany and Microbiology Department, Faculty of Science, Al-Azhar University, Cairo 11884, Egypt
4   Biological and Geological Sciences Department, Faculty of Education, Ain Shams University, Cairo 11566, Egypt; hebaebrahem@edu.asu.edu.eg
*   Correspondence: helbeltagi@kfu.edu.sa (H.S.E.-B.); mahmoud_sofy@azhar.edu.eg (M.R.S.)

**Abstract:** In recent years, nutrient management has gained much attention as a way to mitigate heavy metal stress. Silicon (Si) promotes plant defense responses against toxic metal stresses. In this study, we evaluated the effects of silicon (Si) on copper (Cu) toxicity in two flax genotypes (Sakha 1 and Sakha 2) as it relates to plant growth, yield attributes, total chlorophyll, nucleic acid content, enzymatic and non-enzymatic antioxidants, oxidative damage, lipid peroxidation, copper and silicon content, and fatty acid composition. The results showed that Cu (100 and 200 µM) inhibited plant growth and increased Cu accumulation in soil, roots, and shoots. Cu significantly decreased the yield attributes, total chlorophyll by 9.5% and 22% in Sakha 1 and by 22.5% and 29% in Sakha 2, and enhanced the accumulation of non-enzymatic (tocopherol), enzymatic antioxidants such as superoxide dismnutase, peroxidase, ascorbate peroxidase and catalase) and secondary metabolites (phenol and flavonoids). The DNA content significantly decreased in stressed plants with 100 and 200 µM Cu about 22% and 44%, respectively, in Sakha 1 and about 21.6% and 34.7% in Sakha 2, and RNA content also decreased by about 20% and 29%, respectively, in Sakha 1 and by about 2% and 13% in Sakha 2 compared to the control plant. Furthermore, Cu stress accelerated the generation of reactive oxygen species (ROS), such as hydrogen peroxide ($H_2O_2$) and induced cellular oxidative injury caused by lipid peroxidation. In parallel, Cu induced a change in the composition of fatty acids, resulting in lower unsaturated fatty acid levels and increased saturated fatty acids (increased saturation/unsaturation ratio for both genotypes). Treating the flax plants with irrigation three times with Si protected the plants from Cu toxicity. Si treatment decreased the uptake and the transport of Cu to the shoots and harvested seeds and promoted plant growth, yield attributes, and antioxidant defense systems by reducing Cu accumulation, lipid peroxidation, and the generation of $H_2O_2$. In addition, the alleviation of Cu toxicity correlated with increased Si accumulation in the roots and shoots. In conclusion, Si can be used to improve the resistance of flax plants to Cu toxicity by up-regulating the antioxidant defense system such as superoxide dismutase (SOD), peroxidase (POD), ascorbate peroxidase (APX) and catalase (CAT) and decreasing the oxidative damage caused by reactive oxygen species (ROS).

**Keywords:** copper sulfate; lipid peroxidation; yield; enzymatic and non- enzymatic antioxidants; fatty acids

## 1. Introduction

Heavy metal pollution has become a serious problem worldwide for food safety and the environment due to industrialization, globalization, smelters, foundries, and sludge [1]. High concentrations of heavy metals in agricultural land can have adverse effects on plants, resulting in decreased crop yields and plant productivity [2]. The contamination of agricultural soil involves corruption of the soil that affects its characteristics and its physical, chemical, or biological properties in a negative way that may affect all living things on its surface, including humans, animals, and plants, either directly or indirectly. The contamination of agricultural soil in Egypt depends on the type of pollution, land characteristics, climatic conditions, and the use of pesticides, mineral fertilizers, and reused wastewater in irrigation [3]. In Egypt, the agricultural pollution caused by chemical pesticides produced a loss of 100% [3].

Copper (Cu) is an essential micro-element of plant growth and plays a vital role in several physiological and biochemical higher plant processes, such as antioxidant activity, cell wall metabolism, photosynthesis, the electron transport chain (ETC), mineral nutrient uptake, chlorophyll biosynthesis, and the production of carbohydrates, lipids, proteins, and nucleic acids [4]. Copper pollution is generated by human operations, such as smelting, mining, plating, steel working, refining, household emissions, fertilizer and fungicide application, and sewage sludge [5]. Excess Cu results in intense phytotoxic reactions affecting plant growth and biomass, seed germination, DNA, photosynthesis, pigment production, the metabolism of nitrogen and proteins, and cell membrane mineral absorption [6,7]. Excessive Cu in the food chain may be toxic not only to plants but also to humans and may, therefore, pose a potential threat to human health. Cu can interfere with protein composition, inactivate enzymes by binding to thiols, and replace other essential metals [8,9]. Moreover, excess Cu induces the production of reactive oxygen species (ROS) that become harmful to plant tissues and cause oxidative stress leading to cell death. ROS causes damage to biomolecules such as DNA, proteins, and lipids and serves as a second messenger in intracellular signaling cascades at low/moderate concentration, thereby mediating multiple responses in plant cells. Plants can scavenge ROS by stimulating antioxidant activities via, e.g., superoxide dismutase (SOD) and peroxidase (POD), which also play a significant role in reducing toxicity to Cu in plants such as *Zea mays* L. [9]. In addition, a heavy metal overdose could suppress a plant's antioxidant defense [10,11].

Metal toxicity can be controlled through externally applied nutrient elements, such as Ca and Si. These elements have received much attention around the world in the recent past. Using this strategy, a safe and sufficient nutrient supply at the correct time induces metal stress tolerance capability in crop plants [12,13]. Externally added nutrients both directly (by interacting with toxic metals at the root's surface) and indirectly (by increasing the impact of the accumulation–dilution of biomass) help to overcome toxicity from metals that are stressed by plants [13]. Indirect effects induced by nutrients may also include root character rearrangements, the production of new biosynthetic pathways, oxidative stress metabolism, and the activation of antioxidant defense systems [13].

Silicon (Si) is an important mineral in the soil and is the second most abundant element in the soil after oxygen [14,15]. It plays a key role in plant growth [16], chlorophyll content, and enzyme activity [17], particularly under stressed conditions [14]. Silicon decreases the toxicity of heavy metals by reducing the uptake and accumulation of heavy metals or by altering the metal's composition by introducing a silicon compound, stimulating enzymatic antioxidants, complexing and compartmentalizing silicon with metal ions, regulating cell wall transportation [18,19], and altering the expression of genes involved in Cu homeostasis. For example, the Si supplementation of *Arabidopsis thaliana* under Cu toxicity induced increases in the mRNA expression and enzymatic activity of superoxide dismutase (SOD) [20], an antioxidant enzyme involved in decreasing oxidative stress [20]. Potassium is an important plant nutrient and plays an essential role in a variety of metabolic processes, such as photosynthesis, protein synthesis, and water status maintenance, in plant tissues [21]. Potassium silicate is a source of intensely soluble potassium and silicon and is used primarily as a

source of silicon alteration in agricultural production systems. Small amounts of potassium have also been shown to improve quality of yield [22].

Flax plant *(Linum usitatissimum* L.) is one of the primary global oilseed plants. It is grown in all countries as a dual plant for both fiber and seed oil. The quality of flax seed oil is generally determined by the amount of its essential fatty acids. Flax oil contains the main omega-3 ($\alpha$-linolenic) and omega-6 (linoleic) fatty acids [21]. In terms of nutrition and pharmaceutical quality, linseed is currently essential as a functional food, and its nutritional components include oil, protein, lignin, fiber, minerals, and vitamins [6,23,24]. In respect to cultivars, major variations were found in most flax yield and quality characters. Furthermore, the output of Sakha 1 genotype ranked the first and reported the highest values of main stem diameter, biological and straw yields per faddan [25]. While Sakha 2 received the highest values for the length of the fruiting region, main stem diameter, straw yield per plant, straw yield per faddan and seed yield produced per unit (NUE) for straw yield per faddan [26].

Therefore, the important goals of our study are (1) to determine the effect of Cu toxicity on morphological characteristics, yield attributes, secondary metabolites, oxidative damage, fatty acid composition, and enzymatic and non-enzymatic antioxidants and (2) to evaluate the role of Si in alleviating the adverse effects of Cu toxicity.

## 2. Materials and Methods

### 2.1. Plant Materials

Two flax genotypes (Sahka 1 and Sahka 2) were kindly obtained from the Fiber Crops Research Department, Field Crops Research Institute (F.C.R.I.), Agricultural Research Center (A.R.C.), Egypt. Pedigree and origin of the two flax genotypes under study are presented in Table 1.

**Table 1.** The pedigree of flax genotypes.

| Genotypes | Pedigree | Type | Origin |
|-----------|----------|------|--------|
| **Sakha 1** | Bombay (USA) x I. 1485 (USA) | Dual | Local variety |
| **Sakha 2** | I.2348 (Hungary) x I. Hera (India) | Dual | Local variety |

### 2.2. Experimental Procedures

The experiment was conducted at the Ain Shams University Faculty of Education, Cairo, Egypt. For three minutes, homogeneous seeds were sterilized by a solution of 0.01 M $HgCl_2$ and then washed thoroughly with distilled water. Flax genotypes (Sahka 1 and Sahka 2) were seeded in pots with a diameter of 25 cm containing 3.5 kg of homogeneous clay soil during October 2018. The soil characteristics were as follows: textured sandy loam, 80% sand; 15.5% silt; 4.5% clay; pH 7.8; 0.4 dSm$^{-1}$ EC and 0.45% organic matter. The chemical properties of the soil are explained in Table 2.

**Table 2.** Chemical properties of the soil.

| pH | E.C. dSm$^{-1}$ | Cations meq/L | | | | Anion meq/L | | | |
|----|-----------------|---------------|----|----|----|-------------|-----------------|------------------|-----------------|
| | | Na | K | Ca | Mg | Cl | $SO_4$ | $HCO_3$ | $CO_3$ |
| 7.8 | 0.4 | 2.07 | 0.27 | 1.23 | 0.88 | 4.3 | 0.82 | 0.8 | 0 |

Each treatment had five pots, and in each pot, ten seeds were sown. The pots were divided into three groups.

- The first group was divided into two subgroups:

1. Plants grown in soil irrigated with 400 mL/pot tap water to serve as the control.
2. Plants grown in soil irrigated with 400 mL/pot of 1.5 mM potassium silicate ($K_2SiO_3$) only three times 25, 30, and 35 days from sowing.

- The second group was divided into two subgroups:

1.  Plants grown in soil irrigated with $CuSO_4$ concentrations (100 μM).
2.  Plants grown in soil irrigated with $CuSO_4$ concentrations (100 μM) + and irrigated with 400 mL/pot of potassium silicate ($K_2SiO_3$).

- The third group was divided into two subgroups:

1.  Plants grown in soil irrigated with $CuSO_4$ concentrations (200 μM).
2.  Plants grown in soil irrigated with $CuSO_4$ concentrations (200 μM) + and irrigated with 400 mL/pot of 1.5 mM potassium silicate ($K_2SiO_3$).

Potassium silicate ($K_2SiO_3$) was ($SiO_2$ 35% + $K_2O$ 10%) and was used to irrigate the plants 25, 30, and 35 days after sowing (otherwise, irrigation was done with tap water). The concentration of potassium silicate was chosen according to Reference [27]. Plants were kept under natural conditions (12–14 h photoperiod, 25–27 °C air temperature, and 65–70% relative humidity).

*2.3. Morphological Characteristics and Plant Sample Preparation*

After 60 days of sowing, ten randomized plants were taken for determination of their morphological characteristics in addition to three randomized plants to determine some biochemical measurements of the copper and silicon content in the soil, roots, and shoots. After 160 days of sowing, ten randomized plants were taken from each treatment at harvest to determine their yield attributes and three randomized plants were used to determine the fatty acid composition.

*2.4. Biochemical Studies*

2.4.1. Estimation of Photosynthetic Pigments

Half gram fresh leaves of the two flax genotypes were ground with a mortar and pestle in liquid nitrogen. The samples were treated with acetone (80%), and the mixture was centrifuged at 20 °C for 5 min at 4500 rpm. In order to estimate the photosynthetic pigments, as described in Reference [28].

2.4.2. Estimation of Total Nucleic Acids (DNA and RNA)

DNA and RNA were extracted in the air-dried shoots of the two flax genotypes with tricholoroacetic acid (10% TCA) and then filtreated [29]. To 3 mL aliquots of the filtrate was added 6 mL of diphenylamine reagent (DPAR), 1.5 g diphenylamine was dissolved in 100 mL of acetic acid, and 2.75 mL of concentrated sulfuric acid was added. The reagent was stored at 2 °C, and used within 6 months. The reagent was mixed properly and the tubes heated in boiling water bath for 10 min and left to cool. Using citrate buffer as blank, the absorbance at 595 nm was taken by UV spectrophotometer or by colorimeter [30], while the orcinol reaction was used to estimate the RNA [31]. To 0.6 mL of the acid-hydrolized tissue extract with a RNA content were added 0.4 mL of a stock solution containing 0.5% (*w/v*) cupric acetate in glacial acetic acid, 2 mL distilled water, and, finally, 3 mL of the reagent prepared by dissolving 1 g orcinol in 100 mL concentrated hydrochloric acid. Into two tubes marked Blank and RNA standard were added, respectively, 0.6 mL of a 0.5 N perchloric acid (PCA) solution, and 0.6 mL of the working standard solution containing 10 mg reagent grade RNA in 100 mL of 0.5 N PCA. Into both tubes were then added 0.4 mL of the cupri'c acetate solution, 2 mL distilled water, and 3 mL of the reagent, as added to the test tube. The content of the tubes were mixed thoroughly, and the tubes kept in a vigorously boiling water bath for a period of 10–15 min. The bluish-green color which develops within 5 min. of boiling was read at 660 nm using spectrophotometer.

2.4.3. Determination of Total Phenols

Free phenols were analyzed using a Folin–Ciocalteu reagent and a sodium carbonate solution according to Reference [32]. One hundred milligrams of dried flax leaves were mixed with 6.5 mL

methanol (50%). The samples were then vortexed, allowed to stand for 90 min in the dark at room temperature, and centrifuged at 12,000× $g$ for 5 min. One milliliter of the supernatant was added to 0.8 mL of 25% Folin–Ciocalteu reagent, 5 mL of 85% phosphoric acid, and 10 mL of concentrated hydrochloric acid. The tubes were incubated at 42 °C for 9 min. Then, the absorbance was read at 765 nm using a spectrophotometer.

### 2.4.4. Determination of Total Flavonoids

Using 80% cold methanol, the total flavonoids in the dry flax genotype shoots were extracted and then filtrated. The total flavonoids were determined in the filtrate following a previously reported spectrophotometric method [33]. The methanolic extract filtrate (250 μL) was mixed with 1.25 mL of distilled $H_2O$ and 75 μL of a 5% $NaNO_2$ solution. After 5 min, 150 μL of a 10% $AlCl_3 \cdot H_2O$ solution was added and filtered for 6 min. About 500 μL of 1 M NaOH and 275 μL of distilled $H_2O$ were added to the mixture, mixed well and the intensity of pink color was measured at 510 nm. The level of total flavonoid concentration was calculated using quercetin (QU) as a standard. The results were expressed as mg of quercetin equivalents per 100 g of dry extract.

### 2.4.5. Determination of Tocopherol

500 mg of dry flax genotype shoots were homogenized with a 10 mL mixture of acetone: methanol (7:3, $v/v$) and the mixture was centrifuged for 20 min at 10,000 g. The supernatant was then collected and read by a spectrophotometer (Unicam UV-300A model) at 520 nm [34] based on ferrous -dipyridyl color reagent. Briefly, aliquots of the extracted solution were mixed with 1 mL of 2, 2′-dipyridyl reagent (0.125 g in 25 mL of absolute ethanol then, a one mL portion of ferric chloride reagent (0.2 g in 100 mL of absolute ethanol) was added and the mixture was shaken for 10 sec. The absorbance of the mixture was read at 522 nm in a 1-cm cell 50 sec after adding the ferric chloride. A blank was run, using $CHCl_3$, 2, 2′-dipyridly reagent, and ferric chloride reagent. The absorbance of this solution was measured at 522 nm against a blank. Then, the standard curve was drawn and $\alpha$-tocopherol contents in the extracts were calculated from the regression equation of the standard curve.

### 2.4.6. Determination of the Total Soluble Protein

Using the technique of Bradford [35], crystalline bovine serum albumin was used as a standard to evaluate the protein in the supernatant of the flax genotype shoot enzyme extract.

### 2.4.7. Extraction of Enzymes and ROS Content

Fresh samples of the flax genotype shoots were ground in a mixed mono and diphosphate buffer with pH 7.8 and then centrifuged at 15,000 g for 20 min. The resulting supernatant was used for the following assay of enzyme activity.

### 2.4.8. Determination of Antioxidant Enzymes

The resulting supernatant was used for the following assay of enzyme activity and ROS content. The activity of superoxide dismutase (SOD: EC 1.15.1.1) was calculated by measuring the inhibition of pyrogallol auto-oxidation using the method defined by Reference [36]. The solution (10 mL) was mixed with 3.6 mL distilled water, 0.1 mL enzyme, 5.5 mL phosphate buffer (0.1 M, pH 6.8), and 0.8 mL 3 mm pyrogallol (dissolved in 10 mm HCl). The reduction rate of the pyrogallol was measured with a UV spectrophotometer at 325 nm (Model 6305, Jenway).

Peroxidase (EC 1.11.1.7) was analyzed using a solution containing a 5.8 mL phosphate buffer (0.1 M, pH 6.8), 0.2 mL of the enzyme extract, and 2 mL of the 20 mM $H_2O_2$. After the addition of 2 mL of 20 mM pyrogallol, the increase in the absorbance of the pyrogallol was determined using a UV spectrophotometer (Model 6305, Jenway) for 60 s at 470 nm [37].

The ascorbate peroxidase activity (APX: EC 1.11.1.11) was assayed by the method in Reference [38]. The reaction mixture contained potassium phosphate buffer (0.1 M, pH 6.8), 5 mm ascorbate, 0.5 mM $H_2O_2$, and the enzyme extract, the addition of which started the reaction. The rates were corrected for nonenzymatic oxidation of the ascorbate by the inclusion of a reaction mixture without the enzyme extract. The absorbance was read at 265 nm ($e = 13.7$ mm$^{-1}$ cm$^{-1}$).

Catalase activity (CAT; EC 1.11.1.6) was analyzed by the method in Reference [39]. The reaction mixture at a final volume of 10 mL containing 9.96 mL $H_2O_2$ phosphate buffer (0.1 M, pH 6.8) was added to the reaction mixture to produce a final volume of 10 mL containing 40 μL enzyme extract. CAT activity was determined by the rate change of $H_2O_2$ absorbance over 60 seconds with a UV spectrophotometer at 250 nm.

### 2.4.9. Determination of ROS Content

The $O_2^-$ level was determined according to Reference [40]. The reduction of nitroblue tetrazolium (NBT) to formazan by the superoxide anion was monitored as the absorbance at 490 nm increases using the absorbance coefficient 100 mm$^{-1}$ cm$^{-1}$. The hydroxyl radical (OH•) content was monitored according to the procedure described in [41]. $H_2O_2$ content was determined according to Reference [42].

The lipid peroxidation products, as described by Heath and Packer [43], were measured and quantified in terms of their malondialdehyde (MDA) content. In 2 mL of 0.1% (*w/v*) trichloroacetic acid (TCA), 0.5 g of the flax leaves was homogenized, followed by 20 min of centrifugation at 12,000× *g*. The collected supernatant (1 mL) was mixed with an equivalent TCA volume (10%) containing 0.5% (*w/v*) TBARS, heated at 95 °C for 30 min, and then cooled in ice. Next, the absorption of the supernatants was measured at 532 and 600 nm. After subtracting the non-specific absorbance (600 nm), the concentration of MDA was calculated by its molar extinction coefficient of 155 mm$^{-1}$ cm$^{-1}$.

### 2.4.10. Determination of Minerals

The soil, roots, and shoots of the flax genotypes were dried for 24 h at 70 °C. Using a mortar and pestle, the dried samples were ground into a fine powder. The samples (1 g) were cleaned in crucibles and put in a 600 °C electric oven. Next, 5 mL of 2N HCl was added to the cooled ash specimens for 4 h, dissolved in boiling deionized water, filtered, and then finished at 50 mL. The total concentration of Cu was determined by inductively coupled plasma optical emission spectroscopy.

The Bioaccumulation Factor (BF) was calculated using the procedure reported in reference [44]:

$$BF = \frac{Cu \text{ roots}}{Cu \text{ soil}} \tag{1}$$

where Cu roots and Cu soil are the concentration of Cu in mg kg$^{-1}$ in the flax roots and soil, respectively.

The Translocation Factor (TF) was calculated as the procedure described in reference [45]:

$$TF = \frac{Cu \text{ shoots}}{Cu \text{ roots}} \tag{2}$$

where Cu shoots and Cu roots are the concentrations of Cu in mg kg$^{-1}$ in the flax shoots and roots, respectively. TF >1 represents the active translocation of metals from the root to the shoot. The contents of Si in the root and shoot of the flax plant and soil were determined by the colorimetric molybdenum blue method [46]. Briefly, 0.3 g samples of flax plant were ashed in porcelain crucibles for 3 h at 550 °C. The ash was then dissolved in 1.3% hydrogen fluoride, and the Si concentrations in the solutions were measured by the colorimetric molybdenum blue method at 811 nm with a spectrophotometer.

### 2.4.11. Determination of Fatty Acids by Gas–liquid chromatography (GLC)

The harvested seed oils of the two flax genotypes were saponified according to the method outlined by the A.O.A.C. [47]. The released fatty acids were methylated with benzene: methanol: concentrated

sulfuric acid (10:86:4); the methylation took about one hour at 80–90 °C [48]. The methylated samples were subjected to analysis by a GLC Agilent technologies 6890 N Network GC system Oven. The flame ionization detector (FID) condition was as follows: Initial temp, 50 °C, Initial time, 2 min; Inlet temp, 250 °C; Detector temp, 280 °C; Flame Ionization Detector (FID); Flow, 1.5 mL/min; Column, HP-5 (5% phenyl methyl siloxane) L = 30 m, D = 320 μm; Flame thickness, 0.25 μm; Carrier gas, $N_2$ 30 mL/min; $H_2$, 30 mL/min; and Air, 30 mL/min.

## 2.5. Statistical Analysis

The experimental design was completely randomized, so the statistical analyses were conducted using SPSS (Social Science version 26.00) at a probability level of 0.05 [49]. The quantitative results with a parametric distribution of Levene's test were determined using a independent t test for analysis data at Table 1 and three-way ANOVA with a Tukey Test variance analysis. The confidence interval was set at 95%, and the accepted error margin was set at 5%. Graphs were produced using GraphPad Prism 8. The heat map correlation analysis was estimated to show the statistical relationship between the quantitative parameters.

## 3. Results

### 3.1. Preliminary Study

These two genotypes were selected after preliminary studies on six flax genotypes. We found that the other four genotypes are moderately tolerant to copper stress but that the other two genotypes (Sakha 1 and Sakha 2) are sensitive to copper stress. Thus, we focused on these genotypes to increase their resistance to copper stress using Si (Table 3).

**Table 3.** Preliminary studies to determine the sensitivity of flax genotypes to copper stress. The value is the mean of ten replicates after 10 days of sowing.

| Genotypes | Treatment | Radical Length (cm) | Plumule Length (cm) | Fresh Weight of Seedling (mg) | Dry Weight of Seedling (mg) |
|---|---|---|---|---|---|
| Sakha1 | C | 11.18 ± 0.45[c] | 10.73 ± 0.2[a] | 86.93 ± 0.4[bc] | 13.03 ± 0.5[g] |
|  | 200 μM Cu | 5.35 ± 0.23[e] | 6.41 ± 0.3[g] | 56.33 ± 0.6[e] | 7.3 ± 0.3[h] |
| Sakha 2 | C | 12.07 ± 0.4[abc] | 10.6 ± 0.26[ab] | 88.33 ± 0.3[abc] | 14.13 ± 0.4[def] |
|  | 200 μM Cu | 7.11 ± 0.25[d] | 7.35 ± 0.14[f] | 66.67 ± 0.4[d] | 7.4 ± 0.23[h] |
| Sakha 102 | C | 12.33 ± 0.44[ab] | 10.21 ± 0.3[abcd] | 86.67 ± 0.23[bc] | 15.33 ± 0.54[bc] |
|  | 200 μM Cu | 11.90 ± 0.31[bc] | 9.85 ± 0.32[cde] | 84.33 ± 0.53[c] | 14.57 ± 0.65[cde] |
| Sakha 101 | C | 11.75 ± 0.23[bc] | 10.39 ± 0.23[abc] | 93.67 ± 0.4[a] | 13.93 ± 0.43[ef] |
|  | 200 μM Cu | 11.37 ± 0.41[bc] | 10.27 ± 0.1[abc] | 89.33 ± 0.33[abc] | 13.73 ± 0.44[fg] |
| Giza 7 | C | 12.87 ± 0.2[a] | 9.58 ± 0.2[de] | 88.33 ± 0.23[abc] | 16.43 ± 0.34[a] |
|  | 200 μM Cu | 12.30 ± 0.32[ab] | 9.37 ± 2.4[e] | 84.67 ± 0.4[c] | 15.93 ± 0.5[ab] |
| Giza 8 | C | 11.33 ± 0.41[c] | 10.21 ± 0.23[abcd] | 93 ± 0.3[a] | 15.27 ± 0.34[bc] |
|  | 200 μM Cu | 11.27 ± 0.15[c] | 9.99 ± 0.42[bcde] | 92.33 ± 0.21[ab] | 14.85 ± 0.33[cd] |

Each value is a mean of 10 replications and different superscript letters in each column indicate significant difference (LSD $p \leq 0.05$).

### 3.2. Effect of Silicon and Copper on Morphological Characteristics

The results for shoot and root length, the fresh and dry weight of the shoots, and the roots of the two flax genotypes (Sakha 1 and Sakha 2) are presented in Table 4. These results show that the addition of a Cu concentration to natural soil causes a significant ($p \leq 0.05$) decrease in plant growth and development compared with the control. In addition, all morphological characteristics in the two flax genotypes were significantly increased after the application of Si to the soil compared to the unstressed plants. However, Cu+Si treated plants exhibited greater growth recovery than Cu-treated plants. Overall, the measured growth parameters were reduced in the Cu-treated plants compared to the controls, and Si helped the plants recover from Cu toxicity. An ANOVA two-way analysis

showed there are significant differences between treatments, Cu stress, and genotypes but found a non-significant effect between the two flax genotypes.

### 3.3. Effect of Silicon and Copper on Yield Attributes

Compared with the control, all yield attributes (plant height, technical stem length, fruiting zone length, straw yield/plant, seed yield/plant, and seed index) in the two flax genotypes were significantly decreased by increasing the copper sulfate (Table 5). Moreover, using potassium silicate alone or in combination with copper sulfate significantly increased all the above-mentioned attributes compared to the stressed plants (Table 5).

An ANOVA two-way analysis show there are significant differences between treatments alone, and Cu stress alone, but found a non-significant effect between the two flax genotypes and Interaction (T*S*G).

### 3.4. Effect of Silicon and Copper on Total Chlorophyll, Nucleic Acid, and Secondary Metabolites

The total chlorophyll content in the leaves was also affected after treatment with Cu stress (Figure 1). The results showed that irrigation with Cu in natural soil at various concentrations (100 and 200 μM) decreased the chlorophyll content in the leaves by 9.5% and 22% in Sakha 1 and by 22.5% and 29% in Sakha 2 compared to the plants grown in natural soil without Cu.

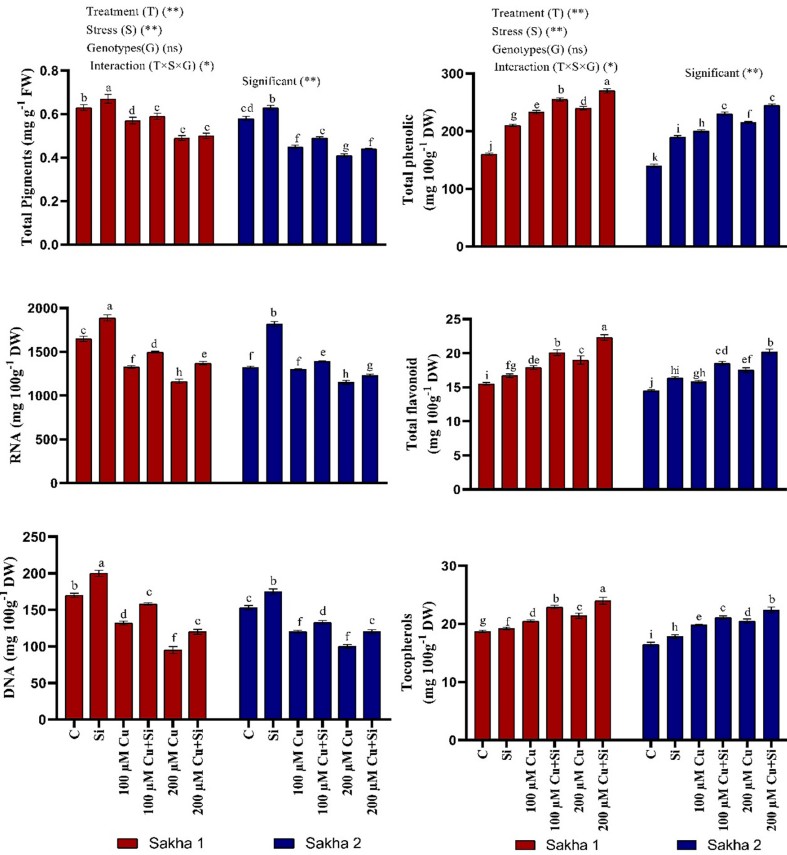

**Figure 1.** Effect of silicon on total chlorophyll in fresh leaves, nucleic acid, and secondary metabolites in the dried shoots of flax plants under copper stress conditions at 60 days old. Each value is a mean (± SE) of 3 replications, and the different letters in each bar indicate a significant difference (LSD *p* ≤ 0.05). * and ** indicate a difference at a *p* value <0.05 or 0.01, respectively; "ns" indicates non-significance.



**Table 4.** Effect of Si on the morphological characteristics of flax plants under Cu stress conditions at 60 days old.

| Genotypes | Treatment | Shoot Length (cm) | Root Length (cm) | Shoot Fresh Weight (g) | Shoot Dry Weight (g) | Root Fresh Weight (g) | Root Dry Weight (g) |
|---|---|---|---|---|---|---|---|
| Sakha 1 | C | 55.8 ± 0.64[b] | 13.4 ± 0.60[b] | 1.50 ± 0.02[b] | 0.225 ± 0.034[b] | 0.150 ± 0.011[c] | 0.051 ± 0.003[c] |
| | Si | 59.8 ± 0.76[a] | 14.9 ± 0.57[a] | 1.56 ± 0.03[a] | 0.268 ± 0.038[a] | 0.191 ± 0.014[a] | 0.062 ± 0.006[a] |
| | 100 μM Cu | 43.6 ± 0.34[g] | 9.3 ± 0.15[g] | 0.60 ± 0.01[g] | 0.116 ± 0.011[def] | 0.118 ± 0.006[de] | 0.031 ± 0.001[f] |
| | 100 μM Cu+ Si | 49.6 ± 0.35[d] | 10.8 ± 0.11[d] | 0.83 ± 0.05[d] | 0.176 ± 0.020[c] | 0.125 ± 0.004[d] | 0.036 ± 0.003[de] |
| | 200 μM Cu | 40.2 ± 0.29[i] | 8.7 ± 0.11[h] | 0.61 ± 0.02[fg] | 0.103 ± 0.005[ef] | 0.103 ± 0.003[ef] | 0.028 ± 0.002[f] |
| | 200 μM Cu+ Si | 45.9 ± 0.38[e] | 10.7 ± 0.12[de] | 0.66 ± 0.03[f] | 0.137 ± 0.017[de] | 0.122 ± 0.002[d] | 0.034 ± 0.003[de] |
| Sakha 2 | C | 50.03 ± 0.46[d] | 10.2 ± 0.19[ef] | 1.17 ± 0.06[c] | 0.143 ± 0.021[cd] | 0.122 ± 0.003[d] | 0.040 ± 0.002[d] |
| | Si | 53.5 ± 0.51[c] | 12.7 ± 0.40[c] | 1.49 ± 0.03[b] | 0.239 ± 0.030[ab] | 0.171 ± 0.007[b] | 0.055 ± 0.002[b] |
| | 100 μM Cu | 39.6 ± 0.22[i] | 8.9 ± 0.08[gh] | 0.58 ± 0.03[g] | 0.103 ± 0.007[ef] | 0.107 ± 0.04[e] | 0.030 ± 0.001[f] |
| | 100 μM Cu+ Si | 44.4 ± 0.21[f] | 11.1 ± 0.34[d] | 0.76 ± 0.04[e] | 0.138 ± 0.015[de] | 0.120 ± 0.03[de] | 0.033 ± 0.002[f] |
| | 200 μM Cu | 36.8 ± 0.29[j] | 8.0 ± 0.17[i] | 0.56 ± 0.01[g] | 0.091 ± 0.008[f] | 0.099 ± 0.001[f] | 0.027 ± 0.002[fg] |
| | 200 μM Cu+ Si | 41.0 ± 0.19[h] | 10.0 ± 0.09[f] | 0.66 ± 0.02[f] | 0.118 ± 0.011[def] | 0.110 ± 0.004[e] | 0.031 ± 0.001[f] |
| Significant | | ** | ** | ** | ** | ** | ** |
| Treatment (T) | | ** | ** | ** | ** | ** | ** |
| Stress (S) | | ** | ** | ** | ** | ** | ** |
| Genotypes (G) | | * | ns | ns | ns | ns | ns |
| Interaction (T*S*G) | | * | ** | ** | ** | ** | ** |

Each value is the mean of 10 replications, and different superscript letters next to the mean values in each column indicate a significant difference (LSD $p \leq 0.05$). * and ** indicate a difference at a $p$ value <0.05 or 0.01, respectively; "ns" indicates non-significance.

**Table 5.** Effect of silicon on the yield attributes of flax plants under copper stress conditions at 160 days old.

| Genotypes | Treatment | Plant Height (cm) | Technical Stem Length (cm) | Fruiting Zone Length (cm) | Straw Yield/Plant (g) | Seed Yield/Plant (g) | Seed Index (g) |
|---|---|---|---|---|---|---|---|
| Sakha 1 | C | 76.5 ± 0.95[b] | 62.2 ± 0.61[b] | 14.3 ± 0.29[d] | 0.24 ± 0.015[b] | 0.222 ± 0.021[c] | 4.00 ± 0.18[bc] |
| | Si | 83.1 ± 1.26[a] | 66.1 ± 0.77[a] | 17. 0 ± 0.37[a] | 0.29 ± 0.017[a] | 0.340 ± 0.025[a] | 5.22 ± 0.31[a] |
| | 100 μM Cu | 67.1 ± 0.88[e] | 53.1 ± 0.50[e] | 14.00 ± 0.18[de] | 0.18 ± 0.010[d] | 0.166 ± 0.013[ef] | 3.02 ± 0.006[fg] |
| | 100 μM Cu + Si | 69.0 ± 0.64[d] | 55.0 ± 0.39[d] | 14.00 ± 0.26[de] | 0.21 ± 0.014[c] | 0.199 ± 0.018[d] | 3.75 ± 0.08[cd] |
| | 200 μM Cu | 63.8 ± 0.73[g] | 50.4 ± 0.46[f] | 13.4 ± 0.13[g] | 0.14 ± 0.012[e] | 0.125 ± 0.008[i] | 2.77 ± 0.011[g] |
| | 200 μM Cu + Si | 65.5 ± 0.60[f] | 52.7 ± 0.31[e] | 12.8 ± 0.15[h] | 0.19 ± 0.01[cd] | 0.156 ± 0.012[fg] | 3.77 ± 0.11[cd] |
| Sakha 2 | C | 72.6 ± 0.51[c] | 59.2 ± 0.51[c] | 13.4 ± 0.25[g] | 0.20 ± 0.011[cd] | 0.188 ± 0.014[de] | 3.70 ± 0.12[d] |
| | Si | 75.5 ± 0.43[b] | 62.5 ± 0.65[b] | 13.0 ± 0.31[e] | 0.24 ± 0.015[b] | 0.267 ± 0.024[b] | 4.11 ± 0.28[b] |
| | 100 μM Cu | 62.3 ± 0.38[h] | 46.3 ± 0.40[h] | 16.0 ± 0.11[b] | 0.15 ± 0.010[e] | 0.130 ± 0.009[h] | 3.22 ± 0.10[ef] |
| | 100 μM Cu + Si | 65.0 ± 0.29[f] | 50.0 ± 0.27[f] | 15.0 ± 0.19[h] | 0.18 ± 0.009[d] | 0.152 ± 0.011[fg] | 3.33 ± 0.15[e] |
| | 200 μM Cu | 58.1 ± 0.59[g] | 44.3 ± 0.35[i] | 13.8 ± 0.08[ef] | 0.10 ± 0.008[f] | 0.105 ± 0.006[j] | 2.25 ± 0.07[h] |
| | 200 μM Cu + Si | 60.0 ± 0.44[i] | 47.5 ± 0.23[g] | 13.5 ± 0.14[fg] | 0.13 ± 0.007[e] | 0.132 ± 0.010[h] | 3.00 ± 0.08[fg] |
| | Significant | ** | ** | ** | ** | ** | ** |
| | Treatment (T) | ** | ** | * | ** | ** | ** |
| | Stress (S) | * | ** | * | * | ** | * |
| | Genotypes (G) | * | * | ns | * | ns | ns |
| | Interaction (T*S*G) | ns | ns | ns | ns | * | ns |

Each value is a mean of 10 replications, and different superscript letters next to the mean values in each column indicate significant difference (LSD $p \leq 0.05$). * and ** indicate a difference at a *p* value <0.05 or 0.01, respectively; "ns" indicates non-significance.

Moreover, treatment with 100 and 200 µM Cu caused a significant decrease in DNA content by about 22% and 44%, respectively, in Sakha 1 and by about 21.6% and 34.7% in Sakha 2, with RNA content decreased by about 20% and 29%, respectively, in Sakha 1 and by about 2% and 13% in Sakha 2 compared to the control plants (Figure 1). In addition, using Si increased the above contents compared to the Cu stressed plants.

The total phenolic, flavonoid, and α tocopherol contents in the shoots of the flax genotypes were significantly increased in response to treatment with Si and Cu alone or in combination compared to the control plants (Figure 1).

### 3.5. Effect of Silicon and Copper on Antioxidant Enzymes

In the present study, the activity of SOD, POD, APX, and CAT enzymes in shoots of flax genotypes was also investigated under Cu toxicity (Figure 2). These results suggested that irrigation with Cu in natural soil increased the activities of antioxidants in the shoots of the two flax genotypes.

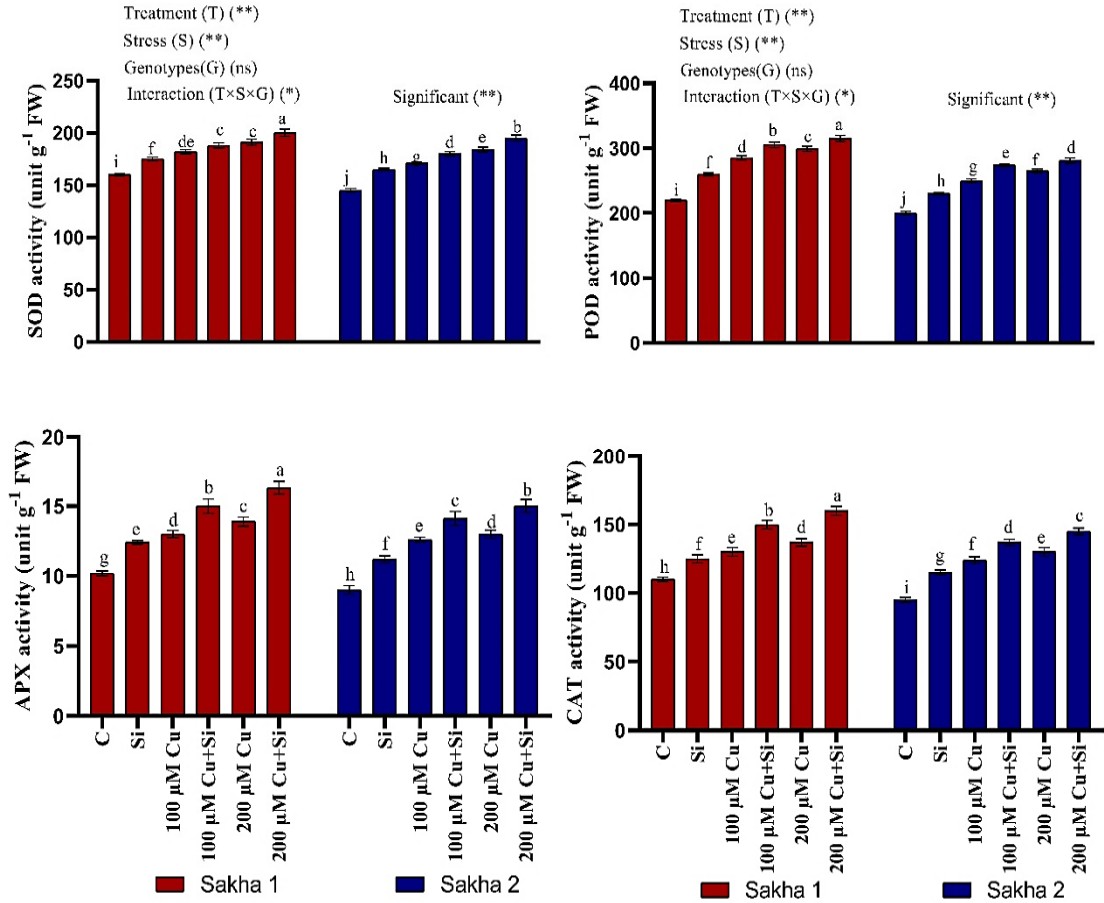

**Figure 2.** Effect of silicon on the antioxidant enzymes of fresh shoots of flax plants under copper stress conditions at 60 days old. Each value is the mean (±SE) of 3 replications, and the different letters in each bar indicate a significant difference (LSD $p \leq 0.05$). * and ** indicate difference at a $p$ value <0.0 or, 0.01, respectively, "ns" indicates non-significance.

### 3.6. Effect of Silicon and Copper on Oxidative Damage

The data in Figure 3 show that the different concentrations of Cu (100 and 200 µM) were significantly increased under oxidative damage (OH, $O_2^-$, $H_2O_2$, and TBARS) in the shoots of flax plants compared to the unstressed plants. In addition, treatment with Si caused a reduction in oxidative damage and the formation of ROS (OH, $O_2^-$, $H_2O_2$, and TBARS) compared to the stressed plants alone.

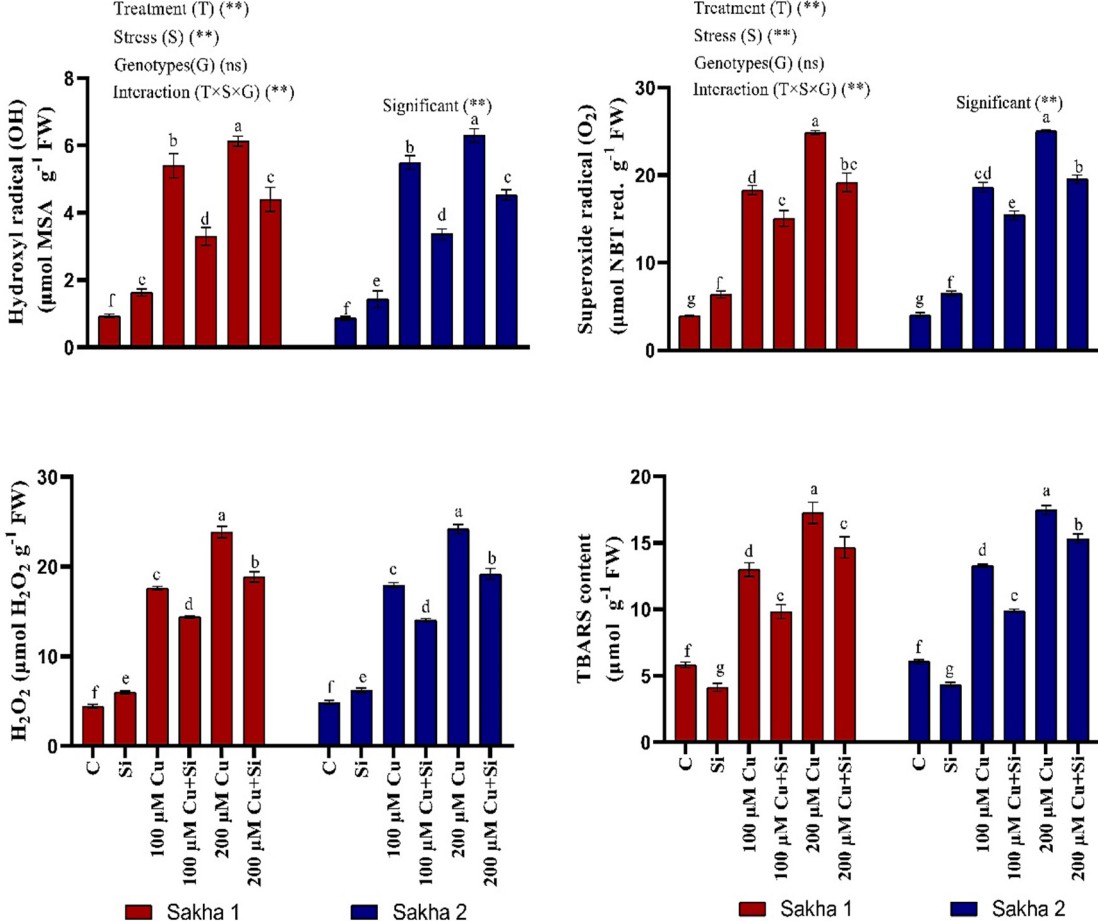

**Figure 3.** Effect of the silicon on ROS content of fresh shoots of flax plants under copper stress conditions at 60 days old. Each value is the mean (±SE) of 3 replications, and the different letters in each bar indicate significant difference (LSD *p* ≤ 0.05). * and ** indicate a difference at *p* value <0.05 or 0.01, respectively; "ns" indicates non-significance.

### 3.7. Effect of Silicon and Copper on Copper Contents

The data in Figure 4 show that the maximum Cu retention was found in the roots followed by the shoots (in descending order) in the two flax genotypes. Treatment with potassium silicate was effective in reducing the plant accumulation of Cu and mitigating the harmful effects of Cu on plant growth, yield, and the physiological properties of flax genotypes. Cu content was significantly increased in the soil, roots, and shoots by increasing the Cu concentrations (100 and 200 μM) compared to the control plants. In addition, treatment with Si caused a reduction in Cu content compared to stressed plants alone.

### 3.8. Effect of Silicon and Copper on Fatty Acid Composition of Harvested Seeds

Oils were derived from the two flax genotypes with six fatty acids, including three saturated (myristic, palmitic, and stearic) and three unsaturated (oleic, linoleic, and linolenic). The saturated fatty acids in the two flax genotypes increased after application with Cu alone or in combination with Si, while unsaturated fatty acids decreased compared to the untreated plants (Figure 5). Cu induced an increased saturation/unsaturation ratio for both genotypes.

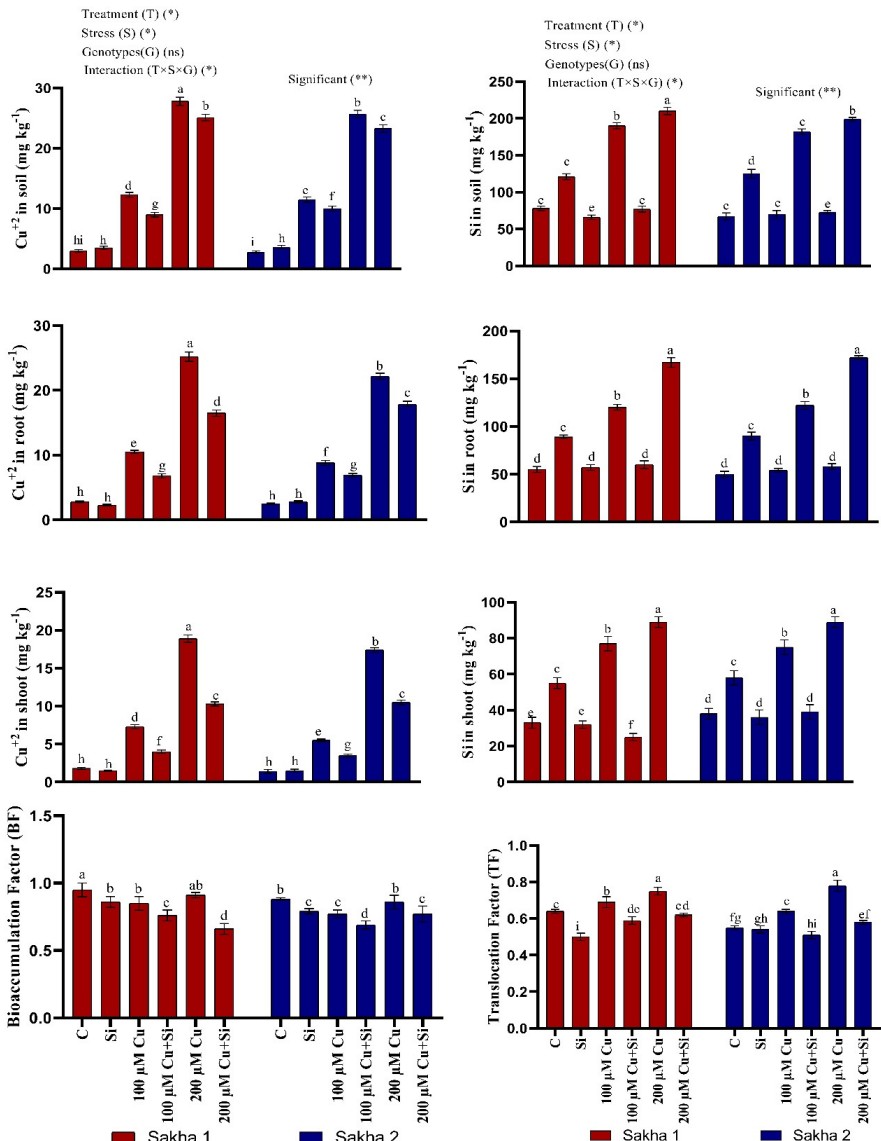

**Figure 4.** Effect of silicon on copper and the silicon content of the dried roots and shoots of flax plants under copper stress conditions at 60 days old. Each value is the mean (±SE) of 3 replications, and the different letters in each bar indicate a significant difference (LSD $p \leq 0.05$). * and ** indicate difference at a *p* value <0.05 or 0.01, respectively; "ns" indicates non-significance.

### 3.9. Correlation

The results of the correlation analysis under Cu stress conditions showed that seed yield/plant and plant height, total photosynthetic pigment, DNA, RNA, total phenolic, flavonoid, α tocopherol, SOD, POD, APX, CAT, Cu (soil, root, and shoot), bioaccumulation factor (BF), translocation factor (TF), myristic, palmitic and stearic, oleic, linoleic and linolenic, saturated, unsaturated, MDA, $H_2O_2$, $O_2$, and OH parameters had a significant correlation with each other (Figure 6). There was a positive and significant correlation between the seed yield/plant and plant height, total photosynthetic pigments, DNA and RNA contents, BF, and oleic parameters. In contrast, there was a negative and significant correlation between seed yield/plant and the total phenolic, flavonoid, α tocopherol, SOD, POD, APX, CAT, Cu (soil, root, and shoot), TF, myristic, palmitic and stearic, linoleic and linolenic, saturated, unsaturated, MDA, $H_2O_2$, $O_2^-$, and OH parameters.

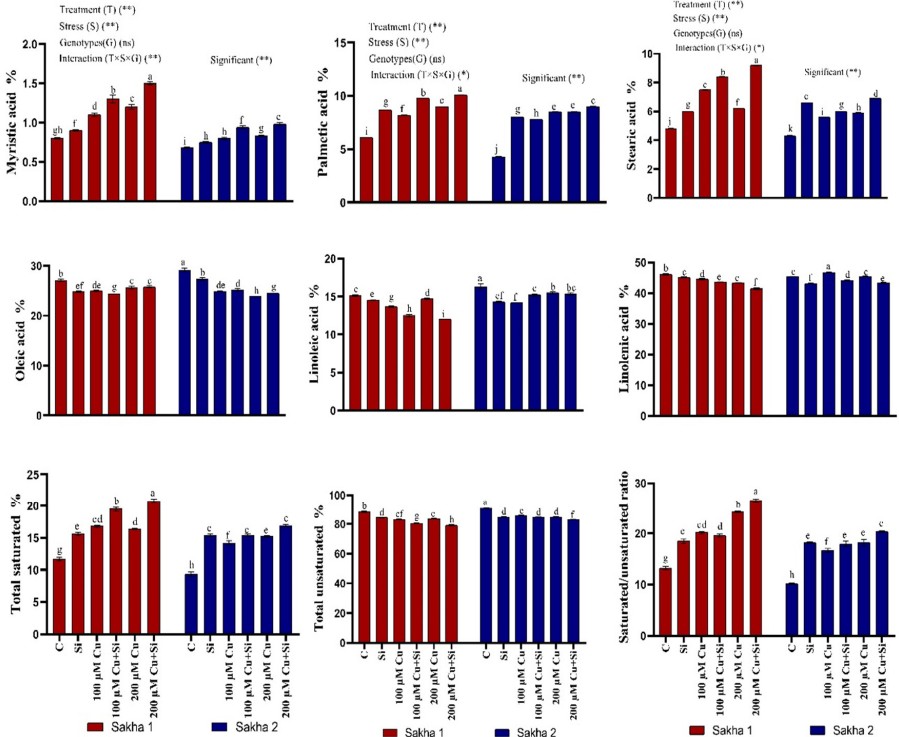

**Figure 5.** The effect of silicon on the fatty acid composition (%) of harvested seeds of flax plants under copper stress conditions at 160 days old. Each value is the mean (±SE) of 3 replications, and the different letters in each bar indicate a significant difference (LSD *p* ≤ 0.05). * and ** indicate difference at a *p* value <0.05 or 0.01, respectively; "ns" indicates non-significance.

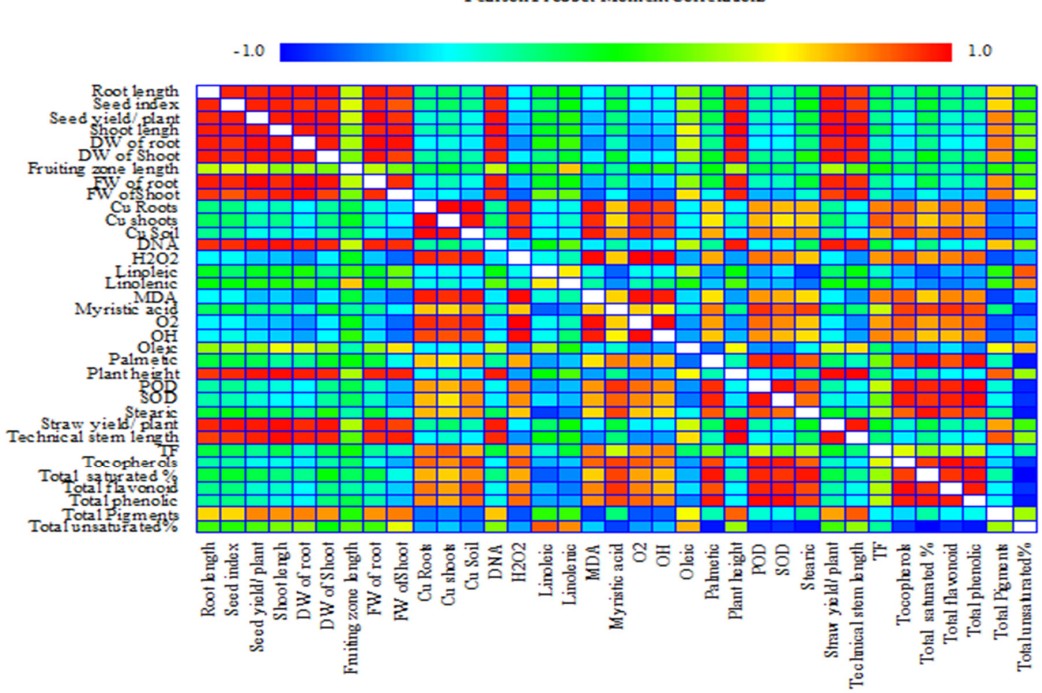

**Figure 6.** Heat map correlation between the seed yield/plant and morphological/biochemical items, along with their statistical significance.

## 4. Discussion

The maximum shoot and root length, fresh and dry weight of shoots, and roots of the two flax genotypes (Sakha1 and Sakha 2) were detected in the plants grown without any addition of Cu in the soil (control), while the addition of 100 and 200 μM Cu to natural soil caused a reduction in the above morphological characteristics. Moreover, using potassium silicate alone or in combination with copper sulfate significantly increased all the above-mentioned attributes compared to the stressed plants (Tables 4 and 5).

This result is consistent with other studies showing that copper stress reduced plant length and fresh biomass in pea plants [27], flax plant [6,23], and also depressed wheat yield [50]. The decline in plant growth was due to a reduction in the absorption of water and nutritional deficiency, which caused a reduction in metabolic activities [51–56]. In rice, Si-rich amendments yielded a reduction in heavy metal accumulation and increased growth in multi-metal (Cd, Zn, Cu, and Pb) contaminated acidic soil [57].

It is known that ROS accumulation induced by copper stress harms nucleic acids, oxidizes proteins, and causes lipid peroxidation [58]. The decline in photosynthetic pigments under copper stress may be attributable to the degradation of thylakoid membranes, the destruction of the chlorophyll apparatus [59], the reduction in chlorophyll synthesis, the protochlorophyll enzyme, and chlorophyllase activity [60].

Treatment with Si enhanced plant growth, chlorophyll, and nucleic acid content under copper stress. The harmful effects of heavy metals on maize plants grown in contaminated soils have been shown to be reduced after treatment with Si [61]. Si plays a key role in promoting plant growth and heavy metal toxicity resistance in rice plants [62]. The mechanisms of Si-mediated heavy metal toxicity alleviation in plants include altering metabolic pathways, enhancing control in heavy metal uptake, stimulating the production of antioxidants, and toxic metal ion immobilization [63]. Furthermore, potassium silicate may increase the photosynthesis of wheat plants and is possibly correlated with enhancing the activity of photosynthetic enzymes, ribulose-bisphosphate carboxylase, $NADP^+$ dependent glyceroldehyde-3-phosphate dehydrogenase, and chlorophyll content under stress conditions [64].

The total phenolic, flavonoid, and α tocopherol contents in the shoots of the flax genotypes were significantly increased in response to treatment with Si and Cu alone or in combination compared to the control plants (Figure 1). Similar results were reported by previous studies [65], which found that the total phenolic compounds increased in *Colobanthus quitensis* after treatment with copper. Phenolics function as soluble ROS scavengers; several authors have defined their roles as antioxidants, such as flavonoids [66]. Phenolic acids are strong antioxidants that resist plant oxidation [67]. Si can mediate detoxification under the stress of Pb, Mn, Zn, and Cu by stimulating non-enzymatic antioxidants [63]. Furthermore, the use of Al in combination with Si in barley cultivars showed a decline in oxidative damage compared to those treated with Al alone [68]. This reduction was followed by an increase in antioxidant ability and the overall concentration of phenol. In this sense, due to the action of Si under different stresses, several studies have observed an increase in antioxidant activity. Consequently, there was an increase in flavonoid content [69,70], antioxidant enzymes [71], gene expression, and activation of the main phenylpropanoid enzymes under heavy metal stress [72]. Further, Si-mediated detoxification of heavy metals primarily requires flavonoid-phenolics or organic acids chelating the metal. Treatment with Si increased production of phenol was observed up to 15 times in maize. Phenolic compounds such as catechin and quercetin have high al-chelating activity which can mitigate Al toxicity in the apoplast root tip [69]. Moreover, the translocation of Cu from root to shoot in wheat was reduced after treatment with Si. This may be due to the increasing proportion of phenolic compounds in the wheat seedlings roots. These studies indicate that Si can indirectly promote heavy metal chelation in plants and thus reduce their phytotoxicity [73].

The activity of SOD, POD, APX, and CAT enzymes in the shoots of flax genotypes (Sakha 1 and Sakha 2) was significantly increased in response to treatment with Si, Cu alone or in combination,

compared to the stressed plants (Figure 2). Similar reports found that antioxidants such as SOD and POD play an important role in scavenging for ROS. Up-regulating the activity of these enzymes demonstrates plants' ability to scavenge excessive ROS in cells. In the soil with a high concentration of Cu, the increasing activity of antioxidants indicated that Linum usitatissimum can tolerate Cu stress by strengthening the antioxidant defense system [23]. The increased SOD and APX activities in rice plants [74] and jack bean were shown to have the same results. Increased CAT activity was also observed in jack bean plants in response to exposure to Cu [73]. As a redox-active transition element, Cu can catalyze ROS overproduction through Haber–Weiss and Fenton reactions [75] with, e.g., superoxide ($O_2{}^-$), hydrogen peroxide ($H_2O_2$), and hydroxyl radicals ($OH^.$). The abilities of the cellular antioxidant system's three enzymes were also studied. SOD enzymes serve as the first line of defense against ROS, catalyzing the disproportion of radicals of $O_2$ to $H_2O_2$ and molecular oxygen; the two essential enzymes for $H_2O_2$ detoxification are CAT and APX. The cell equilibrium between the SOD and $H_2O_2$-scavenging enzymes is considered critical in deciding the steady-state levels of $O_2$ and $H_2O_2$ [76]. Silicon enhances the production of antioxidant enzymes in the growth of plants under heavy metal toxicity [77]. The Si-stimulated enzymatic and non-enzymatic antioxidant processes help reduce oxidative stress by decreasing ROS production. Si-mediated detoxification was also observed under Pb, Mn, Zn, and Cu stress by stimulating enzymatic antioxidants [63]. Several mechanisms have also been proposed which explain the Si-derived benefits to metal detoxification. The generally accepted mechanisms include toxic metal immobilization in soil (at soil level), accumulation of enzymatic and non-enzymatic antioxidants, co-precipitation of metals, phenolic and flavonoid chelation of metal ions, and compartmentation [63].

Several studies have indicated that the oxidative damage generated during environmental stress is due to an imbalance in the production of ROS and antioxidant activity alterations [23]. To avoid the damage caused by oxidative stress, plants have developed many antioxidant systems [75,76]. The Si-stimulated enzymatic and non-enzymatic antioxidant processes help to reduce oxidative stress by decreasing ROS production [63].

Further, Cu and Si content was significantly increased in the soil, roots, and shoot by increasing Cu concentrations (100 and 200 μM) compared to the control plants (Figure 4). There were no significant differences in the silicon content between the two genotypes of flax. Similar results were found, showing that the Si-mediated alleviation of Cu toxicity in the roots of *Nicotiana tabacum* correspond with a two-fold decrease in Cu concentration relative to Cu-treated plants. Thus, the foliar application of Si may be part of a general response and aid in stress alleviation. Silicon may aid plant heavy metal stress tolerance by affecting metal compartmentalization [78].

Treatment with Si caused a marked reduction in the translocation factor (TF), which suggests their ability to immobilize Cu in soil and decrease the transition from soil to plant roots. Likewise, the amount of Cu in the shoot of the *Solanum nigrum* plant was much lower than the amount in the root [79]. The accumulation of Cu in roots with poor translocation to their shoots is correlated with Cu tolerant plants [48]. This response to Cu stresses the immobilization of Cu excess in the root; thus, its exclusion from shooting is an essential mechanism that affords plants Cu tolerance, thereby protecting plants from metal phytotoxicity and allowing them to grow normally [74]. In roots, Si improves metal ion binding by reducing the flow of the apoplastic bypass and decreases toxic metal translocation from roots to shoots. In addition, the accumulation of Si in the cell wall decreased the transpiration rate and thus increased the resistance of the plant to heavy metal stress [64]. The reduction of heavy metal toxicity via the application of Si may be due to the immobilization of soil metals by adsorption, precipitation, complexation or ion exchange, a reduction of soil metal mobility, and transfer to plants [80,81].

Cu induced an increased saturation/unsaturation ratio for both genotypes. Si application was also found to reduce lipid peroxidation and fatty acid desaturation in plant tissues and improve the growth and biomass of plants under heavy metal stress [82]. The unsaturation rate of total fatty acids in maize roots under Cu pressure decreased from 1.42 to 1.28 [83]. A significant positive association was found between both the linoleic acid peroxides and the linolenic acid and the concentration of Cu in the

plantlets [84]. Furthermore, a toxic amount of Cd caused a reduction in the concentration of unsaturated fatty acids and an increase in that of saturated fatty acids in almond seedlings [85]. The degree of the saturation of fatty acids is important to maintain the fluidity of the membrane and provide a correct membrane function environment [86]. An increase in the degree of fatty acid saturation is a normal plant plasma membrane reaction to various stresses in the environment. Additionally, to restore optimum physical membrane properties, some changes in the membrane lipids were considered an adaptive response to different environmental stresses. This seems to be the case in the current research, where certain improvements in the plasma membrane lipids could preserve membrane integrity under heavy metal stress [87].

## 5. Conclusions

In Egypt and all over the world, natural and anthropogenic activities have resulted in a higher concentration of toxic metals in agricultural farmlands, leading to severe adverse effects on crop production and productivity, as well as human health. In this regard, Si has emerged as a practical option to reduce the phytotoxicity and accumulation of toxic metal in plants. The mechanisms by which Si alleviates Cu stress mainly involve the following activities: (1) promoting plant growth chlorophyll content and nucleic acid content, (2) reduction in the root uptake of Cu, (3) protecting against oxidative stress by decreasing MDA and ROS production, and (4) acting directly as an antioxidant to scavenge reactive oxygen species and/or indirectly modulating redox balance through the activation of the antioxidant responses and secondary metabolites accumulation. In conclusion, Si alleviated the adverse effects of copper toxicity and increased the resistance of these genotypes to stress and irrigation by three times, while Si immobilized Cu in the soil and roots and alleviated the oxidative damage caused by ROS.

**Author Contributions:** Conceptualization, H.S.E.-B., M.R.S. and H.I.M.; methodology, H.I.M. and M.R.S.; resources, H.I.M.; formal analysis, M.R.S.; H.I.M. and H.S.E.-B. and validation, H.S.E.-B. and H.I.M.; data curation, H.S.E.-B. and H.I.M.; visualization, H.S.E.-B., H.I.M. and M.I.A.; writing—original draft preparation, H.I.M. and M.R.S.; writing—review and editing, H.I.M., M.R.S.; H.S.E.-B. and M.I.A.; funding acquisition, H.S.E.-B. and M.I.A. All authors have read and agreed to the published version of the manuscript.

**Funding:** This research was supported by the Deanship of Scientific Research, King Faisal University under Nasher track No. 186384.

**Acknowledgments:** The authors thank the MDPI for language editing.The authors would like to thank the Ain Shams University, Faculty of Education, Biological and Geological Sciences and Botany, Microbiology Department, Faculty of Science, Al-Azhar University for promoting this research and would like to thank Deanship of Scientific Research at King Faisal University, Saudi Arabia, for the financial support under Nasher track No. 186384.

**Conflicts of Interest:** The authors have declared that no competing interests exist.

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
