# Peer review of "Silicon Alleviates Copper Toxicity in Flax Plants by Up-Regulating Antioxidant Defense and Secondary Metabolites and Decreasing Oxidative Damage"

_sustainability, doi:10.3390/su12114732_

Round 1

Reviewer 1 Report

The authors in this manuscript have successfully shown that Silicon can prevent copper induced toxicity by inducing the antioxidant defense system in Flax plants. Increasing heavy metal pollution has adversely affected crop yields and plant productivity. Copper is an important micro nutrient and is required for a number of biological processes and plant growth. However, excess copper is toxic not only to the plants but also to humans. Excess copper leads to increased oxidative damage. The authors tested the effect of both copper and silicon on several plant characteristics. The main findings were that copper accumulates in the roots and shoots and leads to decreased plant growth and decreased cholorophyll, nucleic acid and antioxidants. It also induces fatty acid changes, decreasing unsaturated fatty acids and increasing saturated fatty acid content. Silicon treatment improved the growth by decreasing copper absorption and protecting plants from copper induced toxicity.

The authors also used appropriate controls and statistical tests. 

Author Response

Comments and Suggestions for Authors

The authors in this manuscript have successfully shown that Silicon can prevent copper induced toxicity by inducing the antioxidant defense system in Flax plants. Increasing heavy metal pollution has adversely affected crop yields and plant productivity. Copper is an important micro nutrient and is required for a number of biological processes and plant growth. However, excess copper is toxic not only to the plants but also to humans. Excess copper leads to increased oxidative damage. The authors tested the effect of both copper and silicon on several plant characteristics. The main findings were that copper accumulates in the roots and shoots and leads to decreased plant growth and decreased cholorophyll, nucleic acid and antioxidants. It also induces fatty acid changes, decreasing unsaturated fatty acids and increasing saturated fatty acid content. Silicon treatment improved the growth by decreasing copper absorption and protecting plants from copper induced toxicity.

The authors also used appropriate controls and statistical tests. 

Response:

  • Thank you very much for your positive comments and suggestions

  • We Revised our article language editing in MDPI

Reviewer 2 Report

Abstract is very generally, don't has any information about specifically results. Introduction is very good level but I miss there some information about Sahka 1 and Sahka 2 genotypes of flax. In material and methods is necessary write more information about plant material. There is only one sentence. Line 136 must be as half gram not 0.5 g. Estimation of total nucleic acids (DNA and RNA) need more detailed description. Line 145 and 194 authors describe cucumber leaves? Please unify the units in all pages. In results table 1 is genotype Giza and nowhere in text, it is right? If is OK please desrirbe information about this genotype. Statistically differences in table 1 is with * and  in table 2,3 with letter, is not adequate. Figure 7 does not belong to the conclusion.  

Author Response

Reviewer 2

Very thanks to you for your revision and comments

Comments and Suggestions for Authors

1- Abstract is very generally, don't has any information about specifically results.

Response: We add some results specific information and rewrite it

2- Introduction is very good level but I miss there some information about Sahka 1 and Sahka 2 genotypes of flax.

Response: We added some information about Sahka 1 and Sahka 2 genotypes of flax (line 104-108).

3- In material and methods is necessary write more information about plant material. There is only one s entence.

Response: We added it .

Genotypes

Pedigree

Type

Origin

Sakha 1

Bombay (USA) x I. 1485 (USA)

Dual

Local variety

Sakha 2

I.2348 (Hungar) x I. Hera(India)

Dual

Local variety

4- Line 136 must be as half gram not 0.5 g.

Response: Done

5- Estimation of total nucleic acids (DNA and RNA) need more detailed description.

Response: Done line 159- 175

6- Line 145 and 194 authors describe cucumber leaves?

Response: Corrected to flax; thank you very much for this note.

7- Please unify the units in all pages.

Response: Done

8- In results table 1 is genotype Giza and nowhere in text, it is right? If is OK please desrirbe information about this genotype.

Response: these genotypes Giza 7 and 8, sakha 101 and 102 used only in the preliminary study to chose two genotypes that sensitive to Cu stress to increase the resitance to copper stress by silicon treament

These two genotypes were selected after preliminary studies on six flax genotypes. We found that the other four genotypes are moderately tolerant to copper stress but that the other two genotypes (Sakha 1 and Sakha 2) are sensitive to copper stress. Thus, we focused on these genotypes to increase their resistance to copper stress using Si

Statistically differences in table 1 is with * and  in table 2,3 with letter, is not adequate.

Response: modified it

Each value is a mean of 10 replications and different letters in each column indicate significant difference (LSD p ≤ 0.05).

Figure 7 does not belong to the conclusion. 

Response: we delete it

Reviewer 3 Report

The manuscript entitled: “Silicon alleviates copper toxicity in flax plants by up-regulating antioxidant defense and secondary metabolites and decreasing oxidative damage” seems proerly assessed. Results and observations are justfied. It fits the scope and aims of he Journal. The topic is interesting even if it has beeen eplored in the recent literature with reference of Copper (II) toxicity in different vegetal matrices addressed also to human consumption (e.g. crops, etc.). See the recent papers which should be cited:

De Conti et al. Iron fertilization to enhance tolerance mechanisms to copper toxicity of ryegrass plants used as cover crop in vineyards. Chemosphere. 2020, 243,125298. doi: /10.1016/j.chemosphere.2019.125298.

Khorasaninejad et al. Effects of silicon on some phytochemical traits of purple coneflower (Echinacea purpurea L.) under salinity. 2020. Scientia Horticulture, 264, 108954. doi: 10.1016/j.scienta.2019.108954.

Etesami et al., The Use of Silicon in Stressed Agriculture Management. Chapter 19. In: “Metalloids in Plants: Advances and Future Prospects”. 2020. doi: 10.1002/9781119487210.ch19.

In the paper elements are mentioned but the all reasoning it is referred to metal ions: please state this at the beginining. The proposed results are in agreement with previous reported data. Notheless, there is some novelty in the possible explanation which is given given regarding the mechanism of action of silicon in the vegetal matrix evaluated. However, there is also to be considered which copper and silicon ion form is under evaluation during the study and this point cleared.

Moreover, flaxseed oil (lineseed oil) is on the other hand also used as food supplement and care should be given to its potential toxicity if seeds derivatives are going to be used as foodstuff and/or feed supplements. This aspect regarding safety and risk should be mentioned. Linum usitatissimum L. is a plant used for both fiber and seed oil obtaimement, and linseed is currently also considered among the functional foods other than a feed additive. The following papers exploit recent observation in feed area and should be cited:

Omri et al., Effect of a Combination of Fenugreek Seeds, Linseeds, Garlic and Copper Sulfate on Laying Hens Performances, Egg Physical and Chemical Qualities. 2019. Foods. 8, 311, doi:10.3390/foods8080311.

Omri et al., Egg Yolk Antioxidants Profiles: Effect of Diet Supplementation with Linseeds and Tomato-Red Pepper Mixture before and after Storage. Foods. 2019. 8(8). pii: E320. doi: 10.3390/foods8080320.

In the experimental section please give details on how measurements have been done (see e.g. flavonoids, tocoferol, etc.) withouth giving only the Reference.

Please give more details on the experimental pocedures used (see lines 134 and following).

In line 151 it is mentioned Cu+ is this correct?

Conclusion section should be rephrased for clarity stressing the reached end points and novelty of the proposed research. Figure seven could be used as graphical abstract and nota t the end of the Conclusion section of the manuscript.

English language check of the entire manuscript would be useful for better clarity.

Author Response

Reviewer 3

Comments and Suggestions for Authors

1- The manuscript entitled: “Silicon alleviates copper toxicity in flax plants by up-regulating antioxidant defense and secondary metabolites and decreasing oxidative damage” seems proerly assessed. Results and observations are justfied. It fits the scope and aims of the Journal. The topic is interesting even if it has beeen eplored in the recent literature with reference of Copper (II) toxicity in different vegetal matrices addressed also to human consumption (e.g. crops, etc.). See the recent papers which should be cited:

De Conti et al. Iron fertilization to enhance tolerance mechanisms to copper toxicity of ryegrass plants used as cover crop in vineyards. Chemosphere. 2020, 243,125298. doi: /10.1016/j.chemosphere.2019.125298.

Added number 57

Khorasaninejad, S.; Zaire, F.; Hemmati, K. Effects of silicon on some phytochemical traits of purple coneflower (Echinacea purpurea L.) under salinity. Scientia Horticulture, 2020,264, 108954. doi: 10.1016/j.scienta.2019.108954.

Etesami, H.; Jeong,  B. R.; Rizwan, Muhammad. The Use of Silicon in Stressed Agriculture Management. Chapter 19. In: “Metalloids in Plants: Advances and Future Prospects”. 2020. doi: 10.1002/9781119487210.ch19.

Response: Done, the three references cited in our article number 24-26, 70, 81

2- In the paper elements are mentioned but the all reasoning it is referred to metal ions: please state this at the beginining. The proposed results are in agreement with previous reported data. Notheless, there is some novelty in the possible explanation which is given regarding the mechanism of action of silicon in the vegetal matrix evaluated. However, there is also to be considered which copper and silicon ion form is under evaluation during the study and this point cleared.

Moreover, flaxseed oil (lineseed oil) is on the other hand also used as food supplement and care should be given to its potential toxicity if seeds derivatives are going to be used as foodstuff and/or feed supplements. This aspect regarding safety and risk should be mentioned. Linum usitatissimum L. is a plant used for both fiber and seed oil obtaimement, and linseed is currently also considered among the functional foods other than a feed additive. The following papers exploit recent observation in feed area and should be cited:

Omri B.; Manel, B.L.; Jihed, Z.; Durazzo, A.; Lucarini,  M.; Romano, R.; Santini, A.; Abdouli, H. Effect of a Combination of Fenugreek Seeds, Linseeds, Garlic and Copper Sulfate on Laying Hens Performances, Egg Physical and Chemical Qualities. Foods. 2019, 8, 311, doi:10.3390/foods8080311.

Omri et al., Egg Yolk Antioxidants Profiles: Effect of Diet Supplementation with Linseeds and Tomato-Red Pepper Mixture before and after Storage. Foods. 2019. 8(8). pii: E320. doi: 10.3390/foods8080320.

Response: Done, the three references cited in our article number 24-26, 70, 81

3- In the experimental section please give details on how measurements have been done (see e.g. flavonoids, tocoferol, etc.) withouth giving only the Reference.

Please give more details on the experimental pocedures used (see lines 134 and following).

Response: Done

4- In line 151 it is mentioned Cu+ is this correct?

Response: In line 252 we mentioned Cu+Si (It means the treatments Cu and Si together)

5- Conclusion section should be rephrased for clarity stressing the reached end points and novelty of the proposed research.

Response: done and rewrite it

6- Figure seven could be used as graphical abstract and note at the end of the Conclusion section of the manuscript.

Response: we delete it

7- English language check of the entire manuscript would be useful for better clarity.

  • Response: We made language editing in MDPI
  •  

Reviewer 4 Report

In the present study authors showed the role of silicon in alleviating the copper toxicity in flax through analyzing various parameters. The result is presented well, however I am having one major concern about one of the results.

  1. Imposing stress on plant induces activity of antioxidative enzymes such as  SOD, POD, CAT and APX as a part of defensive mechanism. In Figure 2, it has been shown that compared to control plants the level of these enzymes significantly increased by application of only Si. The similar kind of the increased enzyme activity observed due to application of only Cu compared to control. This indicates that Si creating stress to plants, how do you justify these results?

Author Response

Reviewer 4

Comments and Suggestions for Authors

In the present study authors showed the role of silicon in alleviating the copper toxicity in flax through analyzing various parameters. The result is presented well, however I am having one major concern about one of the results.

Very thanks to you of your positive commont

Imposing stress on plant induces activity of antioxidative enzymes such as  SOD, POD, CAT and APX as a part of defensive mechanism. In Figure 2, it has been shown that compared to control plants the level of these enzymes significantly increased by application of only Si. The similar kind of the increased enzyme activity observed due to application of only Cu compared to control. This indicates that Si creating stress to plants, how do you justify these results?.

Response: very thanks to you for your commont

In natural conditions, plants continuously produce several ROS during photosynthesis and respiration processes in cell organelles such as mitochondria, chloroplast, and peroxisomes. Thus, plants can maintain homeostasis by two different detoxification mechanisms involving non-enzymatic and enzymatic antioxidants (Mittler, 2002; Arbona et al., 2003; Apel and Hirt, 2004; Sytar et al., 2013; Wu et al., 2017). So that the amount of antioxidant enzymes increased after treatment with Si alone to help the plant to make defense against any effect. Due to enter Si to the plant, the plant make defence by enhancement the production and accumulation of antioxidants.

The activities of antioxidant enzymes were enhanced with the addition of Si in both normal and copper stress conditions.These results are similar to Manivannan et al., (2016) who found that the activities of antioxidant enzymes were enhanced with the addition of Si in both normal and salinity stress conditions.

 In plants, superoxide dismutase (SOD), catalase (CAT), and ascorbate peroxidase (APX) are the main enzymatic antioxidants, whereas carotenoids, tocopherols, ascorbate, and glutathione are classified as the non-enzymatic antioxidants (Asada, 1999; Racchi, 2013; Kim et al., 2014b,c).

To complete a life cycle, plants are continuously exposed to various abiotic stresses and sometime multifaceted stresses. These stresses in turn causing the generation of various reactive oxygen species (ROS), such as singlet oxygen (1O2), superoxide (O− 2 ), hydrogen peroxide (H2O2), or hydroxyl radicals (OH) in cells (Sharma et al., 2012; Das and Roychoudhury, 2014). These ROS can cause serious oxidative damage to the protein, DNA, and lipids of cell components (Apel and Hirt, 2004; Lobo et al., 2010; Tripathi et al., 2017). Therefore, ROS scavenging is most important defense mechanism to cope with stress condition in plants (Sharma et al., 2012; Baxter et al., 2014; Das and Roychoudhury, 2014). According to previous reports, exogenously Si can improve the ability of ROS scavenging by regulation of antioxidants enzyme activity (Torabi et al., 2015; Kim et al., 2016; Tripathi et al., 2017).

English language check of the entire manuscript would be useful for better clarity.

  • Response: We Revised our article language editing in MDPI
  •  
  • We certify that the following article
  • Silicon alleviate copper toxicity in flax plants by up-regulation of antioxidant defense, secondary metabolites and down-regulation of oxidative damage
  • Hossam S. El-Beltagi *, Mahmoud R. Sofy *, Mohammed I. Aldaej, Heba Mohamed *
  • has undergone English language editing by MDPI. The text has been checked for correct use of grammar and common technical terms, and edited to a level suitable for reporting research in a scholarly journal. MDPI uses experienced, native English speaking editors. Full details of the editing service can be found at â–º https://www.mdpi.com/authors/english.
  • Basel, Switzerland May 2020
  • Shu-Kun Lin Publisher & President MDPI

Round 2

Reviewer 2 Report

The authors accpted all comments. I recommende publish this manuscript.